# Safety and Efficacy of a 48-Month Efinaconazole 10% Solution Treatment/Maintenance Regimen: 24-Month Daily Use Followed by 24-Month Intermittent Use

**DOI:** 10.3390/idr17010007

**Published:** 2025-01-13

**Authors:** Aditya K. Gupta, Elizabeth A. Cooper

**Affiliations:** 1Division of Dermatology, Department of Medicine, Temerty Faculty of Medicine, University of Toronto, Toronto, ON M5S 3H2, Canada; 2Mediprobe Research Inc., London, ON N5X 2P1, Canada; lcooper@mediproberesearch.com

**Keywords:** onychomycosis, tinea unguium, dermatophytosis, antifungal agents, efinaconazole, prophylaxis

## Abstract

Background/Objectives: In an 18- to 24-month Treatment Phase with once-daily efinaconazole 10% solution, subjects with onychomycosis showed an increased rate of cure at Month 24 versus the phase III trials. In order to further improve efficacy, we initiated an extended intermittent efinaconazole Maintenance Phase with use 2–3 times weekly for an additional 24 months from Month 24 to Month 48. These are the first data presented for a 48-month efinaconazole use period. Methods: For patients completing 18–24 months of once-daily efinaconazole, the target great toenail from the Treatment Phase was graded as ‘Clinical Cure’ (≤10% affected area) or ‘No Clinical Cure’ (>10% affected area) at Month 24. Mycological and clinical outcomes were assessed every 4 months from Month 24 to Month 48. There were 35 patients who enrolled in the extension and continued intermittent efinaconazole use to Month 48. Patients with ‘Clinical Cure’ at M24 were reviewed for sustained cure at M48; patients with ‘No Clinical Cure’ were reviewed for development of ‘Cure’ at M48. All patients were reviewed at all visits for adverse events that may be related to efinaconazole use. Results: ‘Clinical Cure’ was found in 6 of 35 enrolled patients at Month 24, and clinical cure status was sustained to Month 48 with intermittent efinaconazole maintenance use. For 29 patients with ‘No Clinical Cure’, 3/29 achieved ‘Clinical Cure’ status at Month 48 with intermittent efinaconazole. Effective Cure and Complete Cure rates improved over the maintenance period to Month 48 in subjects without clinical cure at Month 24. Younger patients showed higher cure rates over the maintenance period, but age group cure differences did not reach statistical significance in this dataset, and 49% of the ≥70-year population had at least a 20% reduction in nail area with maintenance therapy to Month 48. There was only 1 case of possible efinaconazole application site reaction in the Intermittent Maintenance Period to Month 48; prolonged efinaconazole use to Month 48 does not appear to increase the risk of reaction. Efinaconazole use periods are associated with very low positive culture rates in this dataset, including typical contaminant organisms, suggesting efinaconazole presence in the nail plate is providing prophylactic therapy. Conclusions: Intermittent efinaconazole may provide suitable prophylaxis of onychomycosis relapse. Prolonged efinaconazole therapy to Month 48 appears to be safe for all ages and can continue to provide prophylaxis of onychomycosis with Intermittent Maintenance use beyond Month 24 to Month 48.

## 1. Introduction

Onychomycosis is a prevalent, chronic fungal nail infection that is difficult to treat and can significantly impact patient mobility and quality of life. Treatment guidelines for toenail onychomycosis frequently recommend an oral agent such as terbinafine, fluconazole, or fosravuconazole where available, and orals may be used in conjunction with topicals for older patient subsets [1,2]. However, physicians may be reluctant to prescribe these oral products to older patients [2], and restrictions of terbinafine use due to potential for drug interactions and systemic adverse effects such as hepatotoxicity leave only topical products as a first-line option for older patients [1]. Topical product efficacy and safety remain very much a focus for onychomycosis.

Efinaconazole 10% topical solution (Jublia) was approved in 2013 in Canada and in 2014 in the USA for mild to moderate dermatophyte toenail onychomycosis. Efinaconazole 10% solution has low surface tension, low affinity to keratin in the nail plate [3], and accumulates in high concentrations in the nail plate and nail bed after continuous application for 14 days, well above the MIC of dermatophytes that cause onychomycosis [4]. It additionally shows a broad spectrum of action against dermatophytes, yeasts, and non-dermatophyte molds (NDMs) associated with onychomycosis [5]. Despite these good pharmaceutical qualities, efficacy with daily use for 12 months remains relatively low [6], and the potential for relapse of infection after Month 12 is of clinical concern.

We performed a 48-month study to investigate the efficacy and safety of an extension to the standard 12-month, once-daily use of efinaconazole. Enrolled patients began at Day 0 with a Treatment Phase using efinaconazole 10% solution once daily for 18 to 24 months. For patients treated for 24 months: Mycological Cure (MC; negative microscopy and culture) was 66.0% at M12, increasing to 71.7% at M24, and Effective Cure (EC; MC and ≤10% affected nail) was 13.2% at M12, rising to 22.6% at M24 [7].

Subjects completing Month 24 were offered an Intermittent Maintenance Phase from Month 24 to Month 48, continuing use of efinaconazole 2 to 3 times weekly. The 48-month observation in this subject subset allows a significantly long period to observe nail outgrowth and efinaconazole safety and could result in improved efficacy outcomes with the efinaconazole 10% solution. We were particularly interested in whether Intermittent Maintenance from Month 24 to Month 48 could provide suitable prophylaxis and prevent relapse for patients achieving clinical cure to Month 24 with daily efinaconazole. The relative safety of efinaconazole and ease of application, along with its ability to interrupt fungal growth, may make it an ideal option for prophylaxis of fungal infection relapse. This is in addition to the efficacy that topical efinaconazole exhibits when used as daily dosing to treat toenail onychomycosis for 24 months [7].

The use of efinaconazole has not been reported beyond 18 months in other studies in the medical literature to date. The extension of efinaconazole use to Month 48 represents possibly the longest systematic follow-up of topical efinaconazole, and such a duration of systematic observation for onychomycosis is rarely achieved in topical studies. Efinaconazole use is becoming widespread as many patients with onychomycosis have contraindications to systemic therapies, and long-term use is a foreseeable reality for most patients needing topical onychomycosis therapy. The availability of systematic long-term safety data will benefit all physicians and patients when making onychomycosis treatment decisions.

## 2. Patients and Methods

A phase IV single-site Canadian trial was performed and included patients aged 18 years or older with mild to moderate distal lateral subungual onychomycosis (20–50% of the toenail affected) in a great toenail designated as the ‘target’ for evaluation. The study was approved by an independent ethics board and by Health Canada (Health Canada Clinical Trials Database: Control Number 228775). All enrolled patients provided written informed consent for both the Treatment Phase and Intermittent Maintenance Phase of the trial prior to enrolling in the phases.

Eligible patients were enrolled in the long-term, once-daily efinaconazole Treatment Phase with use for up to 24 months. The diagnosis was confirmed visually and by dermatophyte growth in the culture at the screening visit. Patients returned periodically within the first 24 months for visual assessment/photographs of toenail area improvements and had mycology samples taken from the target great toenail to assess the mycological status of the target toenail. The efficacy and safety of this initial 24-month Treatment Phase have been described in detail in another publication and will not be reviewed in depth here [7]. Use of efinaconazole daily for up to 24 months was found to be safe for patients of all ages, with efficacy measures continuing to increase with prolonged use to Month 24.

At Month 24, patients were graded on the clinical outcome of the target toenail as ‘Clinical Cure’ (≤10% affected target toenail area) or ‘No Clinical Cure’ (>10% affected target toenail area). All patients, regardless of clinical outcome, were offered continuation into an Intermittent Maintenance Phase, using efinaconazole two to three times weekly (2–3x weekly) for a further 24 months up to the Month 48 final evaluation.

During the Intermittent Maintenance Phase, all patients choosing to enroll were asked to use efinaconazole 10% solution for 2 to 3 days per week, using a schedule that best suited their time and lifestyle, so long as the use was on non-consecutive days (at least one day of non-application between their use days). As with the Treatment Phase, subjects were asked to continue to refrain from any medicated foot treatments or use of nail polish during the study period. Patients returned for visits at Months 28, 32, 36, 40, 44, and 48. Subject safety was reviewed at all attended visits. All nails were assessed for onychomycosis area, length, and thickness measures at each visit. Area assessment was performed visually by a single-site assessor to minimize evaluator error. Mycology testing was repeated for the target toenail at all visits. Mycology samples were taken by a single staff member experienced in nail collection. All mycology data were assessed by the site laboratory mycology staff.

Outcomes reviewed at Month 48 were as follows: ‘Sustained Clinical Cure’ for patients with ‘Clinical Cure’ at Month 24, and ‘New Clinical Cure’ for patients with ‘No Clinical Cure’ at Month 24. Clinical efficacy in both groups was also reviewed using the standard parameters of Mycological Cure, MC (negative potassium hydroxide ‘KOH’ fluorescence microscopy and negative culture); Effective Cure, EC (≤10% affected area and MC); and Complete Cure, CC (0% affected area and MC). The frequency of any AEs and SAEs, as well as those events possibly or probably related to efinaconazole, were tabulated and reviewed to determine if the events suggest there are long-term safety issues developing with long-term efinaconazole use. A statistical review of proportions was performed using Chi-square testing and a 95% confidence interval. Statistical review of means used *t*-test or ANOVA as applicable, and 95% confidence interval.

## 3. Results

At Month 24, 52 patients completing 18–24 months of efinaconazole daily therapy opted to continue in the Intermittent Maintenance Phase for an additional 24 months. Of the 52 enrolled patients, 35 patients completed efinaconazole 2–3 times weekly up to Month 48, the end of the study (lost to follow-up: 16 patients; Month 48 visit schedule noncompliance: 1 patient).

### 3.1. Efficacy

There were 6 patients considered to be a ‘Clinical Cure’ by affected nail area at Month 24, and 29 patients entered with ‘No Clinical Cure’ status (Table 1). The mean age was similar between the active treatment cure/no cure groups, and *Trichophyton rubrum* is the primary dermatophyte in both groups.

For the six patients with ‘Clinical Cure’, the affected area was 0% for four of the six subjects at Month 24, and the remaining two subjects had ≤10% affected area remaining (2% and 10%, respectively). All six patients sustained a clinical cure to Month 48 with intermittent efinaconazole use (Table 2), and the two patients with ≤10% affected area on entry were able to achieve complete clearance of 0% at Month 48. All six patients were able to maintain or newly achieve Mycological Cure, Effective Cure and Complete Cure by Month 48 with Intermittent Maintenance (Figure 1, Table 2).

Following Intermittent Maintenance to Month 48, three patients with ‘No Clinical Cure’ at Month 24 went on to develop ‘Clinical Cure’ at Month 48 (3/29 = 10.3%; Table 2). Eleven of twenty-nine (37.9%) were able to sustain MC from Month 24 to Month 48 (Table 2). Though MC rates appear to drop in ‘No Clinical Cure’ patients during Intermittent Maintenance, there is an increase in the number of patients achieving EC and CC overall to Month 48 (Figure 1). Representative patient photographs are shown in Figure 2.

The enrolled population in this trial is significantly older than the phase III trial populations, which restricted age to 70 years or less [6]. Patients ≥ 70 years old had a higher mean area of infected nail at the start of Intermittent Maintenance (49.1%, Table 1), and a high proportion of the over-70 group was able to achieve at least 20% improvement in affected area with Intermittent Maintenance, though this did not often translate to ≤10% total affected area qualifying as Clinical Cure (≥20% improvement = 43.7% and 14.3% Clinical Cure, respectively, Table 2). The higher mean area at the start may be a factor in the lower rates of EC and CC versus the other two age groups, but outcomes were not statistically different between age groups at Month 48 (Figure 3).

Of note, all patients completing Intermittent Maintenance but failing to achieve MC to Month 48 were mycology-positive by microscopy only; all cultures were negative at Month 48, including no contaminant organisms. From 495 cultures performed throughout both phases of the trial in patients with ‘No Clinical Cure’ at Month 24, only 8 cultures were positive (8/495, 1.6%, Table 2) at any stage of the study once efinaconazole use was started. Only 2/8 positive cultures were dermatophytes, and such growth was associated with the ‘early’ study phase (M12) or efinaconazole lapse (Table 2). It is suspected that sufficient efinaconazole is being retained in the nail plate with Intermittent Maintenance regimens to prevent any incidental dermatophyte/contaminant growth in culture, and lack of efficacy may result from lack of nail outgrowth rather than lack of efinaconazole effect on fungal organisms.

### 3.2. Safety

During Intermittent Maintenance from Month 24 to Month 48, five mild to moderate non-serious adverse events were reported in four patients, and all events were considered unrelated to efinaconazole 10% solution application (Table 3). For serious adverse events (SAEs), two patients experienced three SAEs. The SAEs were within the normal spectrum for the study population, and none of these events were considered related to efinaconazole use (Table 3). All patients reporting AEs/SAEs were able to complete the study to Month 48. No systemic reactions occurred in association with efinaconazole 10% solution application.

Only one patient (1/35, 2.9%) reported possible treatment-related adverse events during the Intermittent Maintenance Phase (Table 3). The subject experienced intermittent mild to moderate erythema and exfoliation around the application sites, typical of the efinaconazole reaction spectrum reported in the phase III trials [6]. The reaction would resolve with a temporary interruption of efinaconazole. The patient was able to continue intermittent efinaconazole to Month 48.

## 4. Discussion

The data presented here represents the first assessment of long-term efinaconazole 10% solution to 48 months. Data from this study shows that extended use of efinaconazole does not increase the risk of adverse events or reactions for patients of any age. For patients achieving clinical cures, continued use of intermittent efinaconazole may help prevent relapse of infection after achieving clinical cures. For patients not achieving clinical cure to Month 24, continued intermittent efinaconazole for an additional 24 months provides the additional opportunity for cure.

In phase III trials, the mycological (MC) and Complete Cure rates (CC) of the efinaconazole 10% solution were 53–55% and 15–18%, respectively [6,8]. Patients in this phase IV study are more representative of the ‘real-world’ onychomycosis population, and our enrolled population was more severe than the phase III studies. Our population begins with onychomycosis penetrating further into the nail plate of the target toenail (lowest proximal extent) than in the phase III populations, with almost 50% of patients having less than 3 mm of clear nail at daily efinaconazole enrollment [7]. It is expected that increased severity would require an overall longer period of outgrowth/lower cure rate at similar time points relative to less severe populations. We also had a significant number of patients greater than 70 years of age. Increased age is a further burden for nail clearance, as there is a decrease in peripheral circulation and slower outgrowth of toenails [9,10,11]. The patient population and target toenails enrolled in this study would represent a significant clinical challenge for any onychomycosis therapy, but the efinaconazole safety profile suggests the treatment could theoretically be continued much longer without an increased risk to patients.

Older patients represent the population most in need of non-oral antifungal treatment options, and being able to confirm safety for these patients is critical. In this study, there were no systemic adverse events associated with the efinaconazole 10% solution, no drug interactions, and all treatment-related adverse events were localized to the site of action. This long-term study provides evidence that efinaconazole 10% solution may be used safely over 48 months, including for elderly patients. In general, our safety data does not show any increased safety risk for the elderly subset.

There are indications that the low rates of clinical cure may be due to ongoing nail dystrophy and lack of ability to clear the dystrophy, rather than any lack of antifungal activity by efinaconazole. No patients continuing with Intermittent Maintenance showed positive dermatophyte culture at Month 48, regardless of outcome status, and there was no evidence of dermatophyte resistance that would show increased positive culture rates or resurgent dermatophyte culture as the study progressed. The presumption that a toenail can clear itself of infection within 12–24 months after significant fungal infection may be a significant underestimate of the time needed, particularly for patients > 70 years old, and extended use of efinaconazole to Month 48 may still not be sufficiently long to provide reliable outgrowth of treated nails. We also noted that females showed significantly higher cure rates than males to Month 48, with 100% achieving MC and 80% achieving Complete Cure, versus 46.7% and 16.7% of males, respectively. The female population in the study is very small versus the male population, making it difficult to draw strong statistical conclusions with these data, but we would hypothesize that females could have an advantage in clearing dystrophic nails. Some studies have suggested physiological differences in hormones and nail structure between males and females may be advantageous in female onychomycosis clearance [12], and females also may perform nail care more frequently and skillfully than the average male, providing improved potential to reduce the burden of infection while receiving antifungal benefit from efinaconazole. Methods to reduce nail dystrophy and increase nail outgrowth may provide better boosts to topical treatment efficacy [13,14].

Compliance over the long treatment periods was a concern. We did not assess intermittent compliance with any measurable method but relied on patient self-reporting of compliance. Patients continuing in the extended study were, by definition, accepting of a long-term use regimen and may not be representative of a general use population, but we noted particularly that elderly patients using medications daily were often already familiar with techniques to effectively incorporate new medications into their daily schedules. It is likely that patients were not 100% compliant with the treatment at all phases of the research, particularly as the treatment was reduced to 2–3 times weekly. Lower compliance may be reflected in the drop in Mycological Cure. Many patients entering the Intermittent Maintenance had some missed visits in the Month 24 to Month 36 period due to local COVID-19 pandemic closure requirements. Missed visits may have lowered compliance and also concurrently resulted in fewer debridement procedures being performed in this period, leading to increased positive microscopy rates. We are not able to assess the impact of non-compliance or lack of full compliance on the outcomes; however, the continuing low rates of positive culture suggest patients were using sufficient application regimens in most cases to maintain antifungal effects in the nail plate, and our data here likely underestimates the potential for efinaconazole as prophylactic therapy.

There are concerns with long-term efinaconazole use, such as the development of resistance and the cost of the drug regimen. Azole drugs have been associated with the development of antifungal resistance, but we saw no mycological evidence of resistance occurring in the subjects continuing efinaconazole use for up to 48 months, where resistance may show as the growth of a dermatophyte in culture after previous negative cultures. Regarding costs, the increased number of toenails involved as well as the duration of efinaconazole use are primary factors in costing. The Intermittent Maintenance regimen reduces the amount of drug applications needed versus a standard daily therapy, improving the cost of efinaconazole use over the long periods of use required. However, the cost may remain significant and prohibitive over the long term, particularly for patients with multiple nail involvement. Future investigation of alternative intermittent therapy patterns may be useful in developing more cost-effective regimens.

## 5. Conclusions

In summary, this trial demonstrates the safety of efinaconazole 10% solution used over 48 months, starting with daily use for 24 months and continuing intermittently for 24 months. Application appears to remain safe even for elderly patients over the 48-month application schedule. Intermittent Maintenance 2–3x weekly appears to provide suitable prophylaxis for patients showing onychomycosis clinical cure who may wish to prevent relapse. Intermittent use also can provide ongoing reduction in infected nail areas, with incremental improvement in cure rates for patients continuing beyond the standard 12-month treatment regimen of the phase 3 trials. Nail clearance remains elusive, but efinaconazole appears to have ongoing activity in preventing fungal growth throughout this trial, with no sign of resistance developing; ongoing nail dystrophy may also need to be addressed to boost topical cure rates.

## Figures and Tables

**Figure 1 idr-17-00007-f001:**
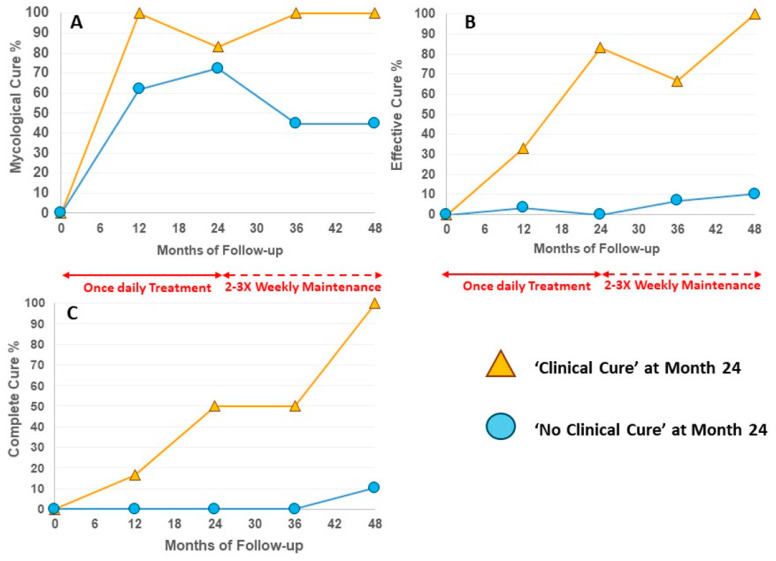
Long-term efficacy of efinaconazole treatment and maintenance. (**A**) Mycological Cure; (**B**) Effective Cure; (**C**) Complete Cure.

**Figure 2 idr-17-00007-f002:**
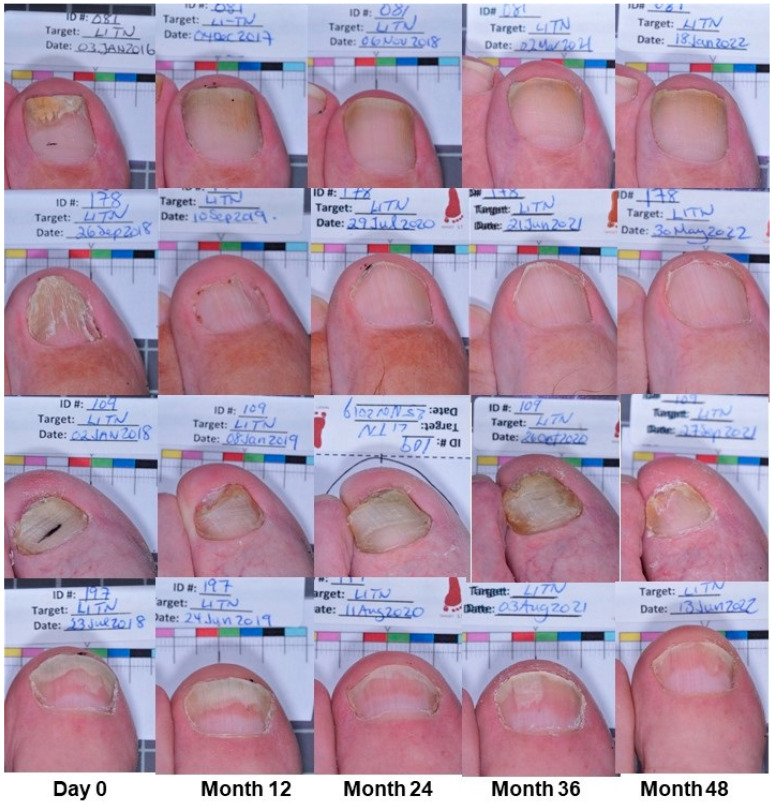
Subject efficacy during efinaconazole treatment/maintenance to Month 48. Photos of target great toenails from four patients from Day 0 to Month 48; extended ‘Cure’ = Subjects 81 (top row) and 178 (second row); Subject 109 (third row, >70 years): significant improvement but no cure; Subject 197 (bottom row): Month 24 intermittent “Clinical Cure”, with relapse at Month 36, recovering cure at Month 48.

**Figure 3 idr-17-00007-f003:**
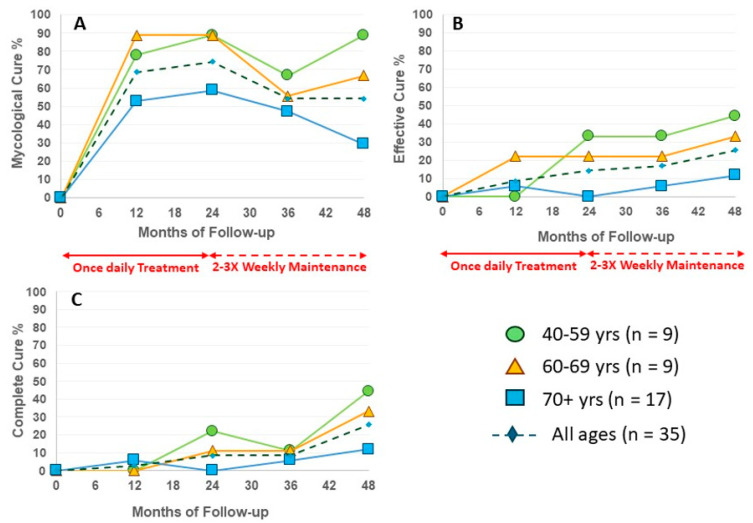
Efficacy of efinaconazole treatment and maintenance by age group. (**A**) Mycological Cure; (**B**) Effective Cure; (**C**) Complete Cure.

**Table 1 idr-17-00007-t001:** Demographics at Month 24: start of Intermittent Maintenance.

**Month 24—All Patients Enrolled to Maintenance**		
**Parameter**	**‘Clinical Cure’ After 18–24 Months Efinaconazole**	**‘No Clinical Cure’ After 18–24 Months Efinaconazole**
N = 52	n = 9	n = 43
Mean Age at Start of Treatment Phase—Day 0, yrs (Min, Max)	57.20 ± 14.01 (24, 70)	68.29 ± 8.30 (47, 82)
Mean % Area Involved after 18–24 months of Efinaconazole, % (Min/Max)	1.89% ± 3.48 (0%, 10%)	43.72% ± 17.39 (15%, 85%)
# of Early Withdrawal Patients Prior to Month 48	Lost to Follow-up: n = 3	Lost to Follow-up: n = 13 Visit Schedule Noncompliance: n = 1
**Patients completing Maintenance Phase Month 48**		
**Parameter**	**‘Clinical Cure’ After 18–24 Months Efinaconazole**	**‘No Clinical Cure’ After 18–24 Months Efinaconazole**
N = 35	n = 6	n = 29
Sex	F = 3 (50%) M = 3 (50%)	F = 2 (6.9%) M = 27 (93.1%)
Mean Age at Start of Treatment Phase—Day 0, yrs (Min, Max)	62.63 ± 7.48 (54, 70)	68.37 ± 8.26 (47, 79)
Age Distribution:	40–59: 3 pts 60–69: 2 pts 70+: 1 pts	40–59: 6 pts 60–69: 7 pts 70+: 16 pts
Mean % Area Involved after 18–24 months of Efinaconazole, % (Min/Max)	2.00% ± 4.00 (0%, 10%)	44.83% ± 18.64 (15%, 85%)
Mean % Area Involved by Age Group:	40–59: 3.3% 60–69: 1.0% 70+: 0%	40–59: 35.8% 60–69: 42.9% 70+: 49.1%
Primary Organism at Start of Treatment Phase:	*T. rubrum*: 5 (83.3%) *T. mentagrophytes*: 1 (16.7%)	*T. rubrum*: 24 (82.8%) *T. mentagrophytes*: 4 (13.8%) *T. tonsurans:* 1 (3.4%)

Treatment Phase = once-daily efinaconazole for 18–24 months; Maintenance Phase = intermittent 2–3x weekly efinaconazole from Month 24 to Month 48; Clinical Cure = less than or equal to 10% infected area in target great toenail plate; F = female; M = male; N = number of patients; *T.* = Trichophyton;

**Table 2 idr-17-00007-t002:** Intermittent Maintenance efficacy to Month 48.

**Parameter**	**‘Clinical Cure’ After 18–24 Months Efinaconazole**	**‘No Clinical Cure’ After 18–24 Months Efinaconazole**
Patients completing Intermittent Maintenance to Month 48	n = 6	n = 29
Primary Outcome	Sustained Clinical Cure with Maintenance to Month 48: 6/6 (100%)	New Clinical Cure with Maintenance to Month 48: 3/29 (10.3%)
≥20% Reduction in Affected Area of Target Toenail to Month 48		Age Group < 70 yrs: 3/13 (23.1%) Age Group ≥ 70 yrs: 7/16 (43.7%)
≥20% Reduction in Affected Area Equaling to ≤10% Total Affected Area at Month 48 (Clinical Cure)		Age Group < 70 yrs: 2/3 (66.7%) Age Group ≥ 70 yrs: 1/7 (14.3%)
Mycological Cure (MC) Status to Month 48 of Maintenance	Sustained MC: 5/6 (83.3%) Gain MC: 1/6 (16.7%)	Sustained MC: 11/29 (37.9%) Gain MC: 2/29 (6.9%) Loss MC: 10/29 (34.5%) Never MC: 6/29 (20.7%)
**Continued Mycological Efficacy: Culture Review**		
**Parameter**	**‘Clinical Cure’ After 18–24 Months Efinaconazole**	**‘No Clinical Cure’ After 18–24 Months Efinaconazole**
Total # Cultures in 48-Month Treatment Period	n = 57	n = 495
Positive Cultures During Efinaconazole Use	3/57 (5.3%)	8/495 (1.6%)
Treatment Phase (Day 0 to Month 24):	1—NDM UNK-likely contaminant (Month 20)	1—*T. rubrum* (Month 12) 1—*Candida* spp. (Month 12) 2—NDM UNK (Month 24)
Maintenance Phase (Month 24 to Month 48):	1—NDM UNK (Month 28) 1—NDM UNK-likely contaminant (Month 40)	1—NDM UNK-possible *Acremonium* (Month 44)
Lapsed Efinaconazole During Pandemic Closures		1—*T. rubrum* (Month 24) 2—NDM UNK-likely contaminant (Month 24)

n = number of patients; Treatment Phase = once-daily efinaconazole for 18–24 months; Maintenance Phase: intermittent 2–3x weekly efinaconazole from Month 24 to Month 48; MC = Mycological Cure, negative microscopy and negative culture; Sustained MC = negative microscopy and culture at the start and end of the maintenance period; Gain MC = change from positive microscopy and/or positive culture at Month 24 to negative microscopy and culture at Month 48; Loss MC = change from negative microscopy and culture at Month 24 to positive microscopy and/or culture at Month 48; Never MC = positive microscopy and/or culture at Month 24 and Month 48; *T.* = Trichophyton; NDM = non-dermatophyte mold; UNK = unknown, NDM organism could not be confirmed with culture morphology exam; likely contaminant indicates an unidentified ‘black mold’ not resembling any of the primary NDMs associated with onychomycosis.

**Table 3 idr-17-00007-t003:** Safety events with efinaconazole 10% Intermittent Maintenance use—Month 24 to Month 48 (2–3x weekly application).

**Adverse Events—Nonserious (n = 5, in 4 patients)**	
	1 pt—2 events: ‘No Clinical Cure’ at Month 24 Mobility impairment secondary to a fall, mild; not related to Study Tx; resolved Itchy sensation, lower legs, mild; not related to Study Tx; ongoing M48
	1 pt: ‘No Clinical Cure’ at Month 24 Dehydration, moderate; not related to Study Tx; resolved; completed to Month 48
	1 pt: ‘No Clinical Cure’ at Month 24 Cyst, L maxilla, moderate; not related to Study Tx; resolved; completed to Month 48
	1 pt: ‘No Clinical Cure’ at Month 24 COVID-19 infection, mild; not related to Study Tx; resolved; completed to Month 48
**Serious Adverse Events (n = 3, in 2 patients)**	
	1pt: M, 73 yr; ‘No Clinical Cure’ at Month 24 Bladder cancer; not related to treatment; surgical treatment, cleared, continued in study to Month 48
	1pt: M, 70 yr; ‘No Clinical Cure’ at Month 24 2 SAEs: recovered from both, continued in study Surgical repair of umbilical hernia; not related to treatment Fractured right leg; not related to treatment Patient continued in study to Month 48
**Treatment-related adverse events: 1/35 patients (2.9%)—Possible recurrent efinaconazole 10% solution reactions during Intermittent Maintenance**	
1 pt: M, 79 yrs; ‘No Clinical Cure’ at Month 24; *T. rubrum*	Subject showed intermittent mild to moderate erythema and exfoliation, typical of previously reported reactions with efinaconazole use. A first brief interval of reaction at Month 36 resolved with a temporary interruption of the drug and did not immediately return. However, the subject found mild to moderate reaction would re-occur occasionally during the next 12 months of study and would resolve again with temporary stoppage of efinaconazole. The subject continued using efinaconazole intermittently with brief interruptions when needed to resolve irritation; continued in study to Month 48 The affected nail area improved but did not achieve clinical or mycological clearance during the extended follow-up.

## Data Availability

The data presented in this study are available on request from the corresponding author due to ethics and patient privacy concerns.

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
