# Peer review of "Safety and Efficacy of a 48-Month Efinaconazole 10% Solution Treatment/Maintenance Regimen: 24-Month Daily Use Followed by 24-Month Intermittent Use"

_2036-7449, 2025, doi:10.3390/idr17010007_

Round 1
Reviewer 1 Report
Comments and Suggestions for Authors
This manuscript by Gupta and Cooper reports a study on efinaconazole, a topical treatment for fungal infections of the nails. The authors extend their original study from 24 months to 48 months. While the manuscript provides valuable insights, there are several issues that must be addressed before it can be considered for publication. The long follow-up time is a strength of the study, and the manuscript could eventually be publishable if the authors address the following points:
1. The authors use "efficacy" and "effective" interchangeably in the title, objectives, and conclusion of the abstract, as well as in the main text. As the authors are likely aware, "efficacy" and "effectiveness" are distinct concepts and should not be conflated. This study, by design, does not assess the "effectiveness" of the drug. For example, there is no data to ascertain whether the six patients enrolled in the extension phase and those who did not enroll (among the patients who achieved Clinical Cure at 24 months) are comparable in terms of effectiveness. Therefore, I strongly recommend revising the manuscript to use terminology that accurately reflects the study's results. For instance, the term "effective" should not be used in the abstract.
2. Oral treatments for onychomycosis are now considered to be safe even in older patients (PMID: 38700273) and are widely used in real-world settings (PMID: 39115330). Thus, the statements in the Introduction (page 2) appear overstated. While I agree that topical treatments can be a suitable alternative for certain patients with concerns about drug interactions, the authors should revise the introduction. Revising the introduction based on this comment will enhance the credibility of the manuscript.
3. Table 1: Data for the 17 patients who did not enroll in the extension study should also be presented in Table 1 or as supplementary data. Providing this information would allow readers to assess the similarities or differences between enrolled and excluded patients, offering a more comprehensive understanding of the study population.
4. Table 2: The data labeled "Total # cultures after efinaconazole start" in Table 2 is unclear. Does this represent numbers counted from the initial study at 0 months to 48 months, or does it reflect results from 24 months to 48 months? Please clarify this point, either directly in the table or in a footnote.
5. On Page 7, the authors introduce information about patients who withdrew from the study. However, this population is not clearly described in the Methods section. If the authors wish to report and discuss these patients, the Methods section must be revised to include details about this population and how their data were handled.
6. The authors claim that treatment effects are lower and take longer to achieve cure in older patients. However, according to Table 2, the proportion of patients aged ≥70 years achieving ≥20% reduction in affected fungal infection area (49.1%) is higher than that of all 29 patients combined (34.5%). This discrepancy between the results and the discussion needs to be clarified, and, if necessary, the discussion should be revised.
7. As a clinician, I have concerns about potential resistance to efinaconazole with long-term use. Has there been any discussion in previous studies about resistance, or is resistance not a significant issue for long-term topical treatment with efinaconazole? Including such information would strengthen the manuscript.
8. In the extension phase, more than half of the 29 patients had positive microscopy and/or culture results, which may suggest the limited efficacy of efinaconazole. The authors should comment on the pros and cons of topical treatment, possibly also including costs as a factor.
Author Response
Thank you for your thoughtful comments. You have raised many important issues, and we have noted some omissions in the data initially presented. We have made many changes to address your concerns and feel the manuscript has now been substantially improved. Please see below our specific responses to your review points.
- The authors use "efficacy" and "effective" interchangeably in the title, objectives, and conclusion of the abstract, as well as in the main text. As the authors are likely aware, "efficacy" and "effectiveness" are distinct concepts and should not be conflated. This study, by design, does not assess the "effectiveness" of the drug. For example, there is no data to ascertain whether the six patients enrolled in the extension phase and those who did not enroll (among the patients who achieved Clinical Cure at 24 months) are comparable in terms of effectiveness. Therefore, I strongly recommend revising the manuscript to use terminology that accurately reflects the study's results. For instance, the term "effective" should not be used in the abstract.
-
- A majority of the uses of the term ‘Effective’ in this manuscript is to discuss the outcome parameter called ‘Effective Cure’; this is standard terminology used across onychomycosis studies, and was not originated by the authors. We continue to use this term and the abbreviation EC because this is standard to the field, and needs to be maintained to allow comparability to outcomes in other published works.
- We have revised statements where ‘effective’ was not used in the context of Effective Cure parameter assessment.
- Oral treatments for onychomycosis are now considered to be safe even in older patients (PMID: 38700273) and are widely used in real-world settings (PMID: 39115330). Thus, the statements in the Introduction (page 2) appear overstated. While I agree that topical treatments can be a suitable alternative for certain patients with concerns about drug interactions, the authors should revise the introduction. Revising the introduction based on this comment will enhance the credibility of the manuscript.
-
- Many physicians remain cautious where oral therapy in the elderly is concerned, despite guidelines that recommend oral therapy as the primary treatment for onychomycosis. North American guidelines further indicate that oral antifungal use is contraindicated with many concomitant medications frequently used for conditions of the elderly.
- We have updated the statements and references in this section (pg1/2) to provide more emphasis on the guidelines recommending first-line oral antifungals for onychomycosis, but have retained some of the precautionary statements with regards to such use.
- Table 1: Data for the 17 patients who did not enroll in the extension study should also be presented in Table 1 or as supplementary data. Providing this information would allow readers to assess the similarities or differences between enrolled and excluded patients, offering a more comprehensive understanding of the study population.
-
- These 17 patients did enroll into the extension study, but 16/17 were lost to follow-up during the maintenance period. One subject was terminated early from study due to visit non-compliance. Data for these subjects has now been added to Table 1 for a more complete comparison with the patient population completing the protocol.
- Table 2: The data labeled "Total # cultures after efinaconazole start" in Table 2 is unclear. Does this represent numbers counted from the initial study at 0 months to 48 months, or does it reflect results from 24 months to 48 months? Please clarify this point, either directly in the table or in a footnote.
-
- This data is for the entire period: Day 0 to Month 48. Row titles in Table 2 have been updated to clarify the periods presented.
- On Page 7, the authors introduce information about patients who withdrew from the study. However, this population is not clearly described in the Methods section. If the authors wish to report and discuss these patients, the Methods section must be revised to include details about this population and how their data were handled.
-
- We do not feel these 2 patients are essential to the paper discussion, and have removed reference to this data. This will allow the manuscript to remain focused only on those patients using active maintenance therapy to Month 48.
- The authors claim that treatment effects are lower and take longer to achieve cure in older patients. However, according to Table 2, the proportion of patients aged ≥70 years achieving ≥20% reduction in affected fungal infection area (49.1%) is higher than that of all 29 patients combined (34.5%). This discrepancy between the results and the discussion needs to be clarified, and, if necessary, the discussion should be revised.
-
- Data as initially presented did not clearly identify that 20% reduction in area did not necessarily correlate to treatment success, where ≤10% total affected area remaining would be a treatment success. Data has been added to Table 2 and the row titles rewritten to better clarify the reductions by age versus % total area remaining outcome by age. Text in lines 167-171 also update the findings. The proportion of older patients achieving reduction is encouraging, but is not indicative of this group having better outcomes per the standard treatment success measures of onychomycosis compared to age group <70yrs.
- As a clinician, I have concerns about potential resistance to efinaconazole with long-term use. Has there been any discussion in previous studies about resistance, or is resistance not a significant issue for long-term topical treatment with efinaconazole? Including such information would strengthen the manuscript.
-
- There is little data about efinaconazole resistance in the medical literature at present. In vitro data is scarce, and provides conflicting information as to the resistance potential of efinaconazole. It remains to be seen if this will become a clinical problem. As an azole, we certainly would suspect there is some potential for resistance development, but did not see any in this data population. We have added some comments about resistance potential in the discussion (paragraph lines 273-277) for readers who may also have this question.
- In the extension phase, more than half of the 29 patients had positive microscopy and/or culture results, which may suggest the limited efficacy of efinaconazole. The authors should comment on the pros and cons of topical treatment, possibly also including costs as a factor.
-
- Per our discussion, though efficacy is low, we do not feel this is necessarily due to efinaconazole being limited in action, as exemplified by the lack of contaminants or causal fungi growing in culture during the trial period. We have added some sentences regarding compliance concerns over long-term topical use (lines 256-261), as well as some of the factors of costing (lines 279-284) which is of course a major concern for patients and physicians.
- We do not feel a full discussion of topical pros and cons is appropriate here, as it is discussed much more thoroughly in other papers and would prevent focus on the data presented.
Reviewer 2 Report
Comments and Suggestions for Authors
Dear author,
First of all, I congratulate you for conducting this highly scientific Phase IV clinical trial on the efficacy and safety of topical efinaconazole in onychomycosis. This manuscript is of clinical relevance, due to the constant challenge of searching for the ideal antifungal. Therefore, I believe it should be published in the journal but first I would like you to consider some suggestions in order to improve readers' understanding.
1) I think the discussion should highlight the problem of the very long treatment period (24 + 24 months) which may lead to non-adherence in patients, as is the case with other topical antifungal treatments.
2) At the end of the manuscript, the authors should reflect on what future research is needed for this drug (efinaconazole) to be approved in other countries, mainly in European countries such as Spain, where it is not currently marketed.
Sincerely.
Author Response
Thank you for the kind comments, and your attention to this manuscript. We have given your comments much thought, as the points raised are very important to onychomycosis. Please see below our responses.
1) I think the discussion should highlight the problem of the very long treatment period (24 + 24 months) which may lead to non-adherence in patients, as is the case with other topical antifungal treatments.
- Compliance is certainly a concern for long-term use. We have added some additional discussion on the compliance issue (lines 256-261).
2) At the end of the manuscript, the authors should reflect on what future research is needed for this drug (efinaconazole) to be approved in other countries, mainly in European countries such as Spain, where it is not currently marketed.
- This is a difficult issue to cover briefly. We feel the scientific data for efinaconazole has been sufficient for some time to allow use in other countries, and it is our understanding that there is no scientific basis for lack of availability in other countries. We have seen reports of an issue with coordination of the regulatory processes needing resolution for European approval. The availability of other suitable topical agents outside of North America has perhaps led to a lack of will/financing to proceed with efinaconazole regulatory approvals to date.
- Our data presented here may be helpful in encouraging efinaconazole use, but is not required for any use approval. There are many aspects of topical onychomycosis therapy requiring more research to improve treatment options, and we do not feel it is appropriate to add such data to the manuscript at this time. We have added however a brief statement about future research on alternate intermittent efinaconazole regimens that may improve costing of long-term use (lines 283-284) which is also of concern to all efinaconazole users/prescribers, and the potential need to take more action against nail dystrophy (lines 254-255; 294-295).
Round 2
Reviewer 1 Report
Comments and Suggestions for Authors
The authors have thoroughly revised the manuscript based on comments in previous round of review or replied appropriately. I believe that the presentation of the methods, results (including tables and legends), discussion has been improved and the manuscript is suitable for this journal. Thank you for the opportunity to review this manuscript.